# Assessment of complementary health approaches use in pediatric oncology: Modification and preliminary validation of the "Which Health Approaches and Treatments Are You Using?" (WHAT) questionnaires

**Mohammad R. Alqudimat**[1,2]*, **Karine Toupin April**[3], **Lindsay Jibb**[1,2], **Charles Victor**[4], **Paul C. Nathan**[5], **Jennifer Stinson**[1,2]

**1** Lawrence S. Bloomberg Faculty of Nursing, University of Toronto, Toronto, Ontario, Canada, **2** Child Health Evaluation Sciences, Research Institute, The Hospital for Sick Children, Toronto, Ontario, Canada, **3** School of Rehabilitation Sciences and Department of Pediatrics, Children's Hospital of Eastern Ontario Research Institute and Institute du Savoir Montfort, University of Ottawa, Ottawa, Ontario, Canada, **4** Institute of Health Policy, Management and Evaluation, University of Toronto, Toronto, Ontario, Canada, **5** Division of Haematology/Oncology, The Hospital for Sick Children, Toronto, Ontario, Canada

* m.alqudimat@mail.utoronto.ca

## Abstract

### Objective

Complementary Health Approaches (CHA) are commonly used by children with cancer; however, a few health care providers (HCPs) inquire about the use of CHA. A standardized questionnaire could facilitate such clinical discussions. We aimed to adapt and determine the face and content validity of the "Which Health Approaches and Treatments are you using?" (WHAT) child and parent-report questionnaires in pediatric oncology.

### Methods

An electronic Delphi survey that included children with cancer (8–18 years), parents, and HCPs and CHA researchers was conducted to reach consensus on the content of the WHAT questionnaires in pediatric oncology. Children and parents from the Hospital for Sick Children (SickKids), and HCPs and researchers from the International Society of Pediatric Oncology and Pediatric Complementary and Alternative Medicine Research and Education Network completed the survey. To determine the face and content validity of the questionnaires, two iterative cycles of individual interviews were conducted with purposive samples of children (8–18 years), parents, and HCPs from SickKids.

### Results

Consensus was reached on all domains and items of the original WHAT questionnaires after one Delphi cycle (n = 61). For face and content validity testing, the first cycle of interviews (n = 19) revealed that the questionnaires were mostly comprehensive and relevant.

**Data Availability Statement:** All relevant data are within the manuscript and its Supporting information files.

**Funding:** Phases 1 and 2 were funded by the Pediatric Group of Ontario (POGO). The funders had no role in study design, data collection and analysis, decision to publish, or preparation of the manuscript.

**Competing interests:** The authors have declared that no competing interests exist.

However, the paper-based format of the original WHAT was not user-friendly, and generic items were vague and not aimed at facilitating clinical dialogues about CHA use. The WHAT questionnaires were then modified into electronic cancer-specific self- and proxy-report questionnaires including 13 and 15 items, respectively. The second cycle (n = 21) showed no need for further changes.

## Conclusions

The modified electronic cancer-specific WHAT questionnaires showed adequate face and content validity. The next step is to determine inter-rater reliability, construct validity, and feasibility of administration of the modified WHAT questionnaires in pediatric oncology.

## Introduction

Complementary health approaches (CHA) encompass health care products and practices that exist outside the mainstream healthcare system (i.e., biomedicine or conventional medicine), and include nutritional (e.g., dietary supplements and herbs), psychological (e.g., mindfulness and spiritual practices), and physical (e.g., massage and spinal manipulation) approaches, or combinations of these approaches (e.g., yoga and acupuncture) [1]. There is a lack of agreement in how to define and classify CHA [2–5]. One of the consequences of this variation in defining CHA is the inconsistency in how researchers measure and describe the patterns of CHA use [2, 3], which has contributed to the variation of the reported prevalence rates [3, 6–8]. The reported prevalence rate of CHA use in pediatric oncology ranges from 6% to 100% (median = 57.8%, n = 7,219 from 34 countries) [7].

The widespread use of CHA by children with cancer generates ethical, legal, and health-related concerns. Children with cancer usually use CHA in conjunction with conventional cancer treatments [6, 7, 9, 10]. This may be beneficial to alleviate cancer symptoms and side effects of cancer therapies; especially if the use is discussed with HCPs and supported by available evidence or clinical expertise. For instance, research has shown that acupuncture would be safe and effective for chemotherapy-induced nausea and vomiting [11–13]. Thus, HCPs should discuss these safe and effective CHA approaches with patients and families as they would discuss any other conventional treatments [14, 15]. Such discussions would help patients and families make informed (i.e., active or collaborative) decisions about using these CHA modalities [14, 16, 17]. However, assessment and discussion of CHA is not routine practice by pediatric oncology HCPs, which may be explained by their lack of knowledge about CHA [17–19], and their worries of potential conventional medicine abandonment as a result of CHA use for cancer care [20, 21]. Research shows that CHA use may delay a cancer diagnosis, and may lead to delayed receipt of or abandonment of conventional cancer treatments [10, 22–26], which may lead to significantly worse survival outcomes [20, 21]. Moreover, using some types of CHA (e.g., natural health products) may lead to a potential risk of interaction with conventional treatments leading to either greater toxicities or lower efficacy of cancer-targeted agents [27–30]. Thus, failure to disclose the use of such products can lead to preventable morbidity or even mortality.

The disclosure rates of CHA use by children with cancer are low in most countries ranging from 8% to 78% (median = 43%) based on data reported from 22 countries [7]. Children with cancer and their parents/caregivers may not voluntarily report information about CHA use unless HCPs explicitly ask [18, 31–36]. Moreover, children with cancer and their parents may

feel hesitant to share this information unless HCPs show acceptance and openness to discuss the use of CHA [34–39]. Fear of doctor's reaction to CHA use was reported in six studies as one of the reasons for not disclosing CHA use by children with cancer and their parents [10, 32, 40–43]. Therefore, it is crucial that HCPs be aware of CHA use, understand its use and its impact on the child's health, and be prepared to initiate an open and non-judgmental discussions about CHA use. These discussions could ideally be facilitated by a questionnaire-based approach [44].

Our team published a recent systematic review of CHA questionnaires in pediatrics, and 35 questionnaires were identified [3]. None of these questionnaires were thoroughly validated, but the "Which Health Approaches and Treatments are you using?" (WHAT) questionnaires (versions for both children and parents) had evidence of initial validation for children with idiopathic arthritis (age 8–18 years) [44]. The WHAT were developed as paper-based short generic questionnaires. Stemming from this finding, we aimed to adapt and validate the WHAT questionnaires for use by oncology HCPs to facilitate open clinical discussions about CHA use with pediatric oncology patients, parents/caregivers, and their families; and assess CHA use by children with cancer 8–18 years of age (i.e., descriptive purpose). The final versions of the WHAT questionnaires must be relevant, comprehensive, and comprehensible but also easy to use by children with cancer (i.e., child self-report) and their parents (i.e., parent proxy-report) in clinical settings.

## Materials and methods

An adapted version of the Behavioural Model of Health Services Use [45] was used to conceptualize the underlying relationship between CHA use by children with cancer and the associated variables, according to the evidence (S1 Appendix). Both the child self- and parent proxy-report versions of the original WHAT questionnaires include four domains: child's CHA use, variables associated with CHA use, perceived impact of CHA use, and communication about CHA within the family and with HCPs. The child self- and parent proxy-reports consist of 13 and 15 items, respectively [44].

### Design

In adapting the WHAT questionnaires, a Delphi survey was conducted (i.e., Phase 1). This survey was distributed electronically to children with cancer, parents, pediatric oncology HCPs, pediatric complementary HCPs, and CHA researchers. Phase 1 allowed consensus and adaption of the domains and items for use in pediatric oncology. Phase 2 explored the face and content validity of the questionnaires, assessing relevance, comprehensiveness, and comprehensibility in the context of pediatric oncology, according to the COnsensus-based Standards for the selection of health Measurement INstruments (COSMIN) guideline [46]. Iterative cycles of individual semi-structured interviews with children with cancer, their parents, and pediatric oncology HCPs were conducted using think-aloud and probing techniques. Interviews focused on identifying potential sources of response error by clarifying the item meaning (i.e., comprehensibility) and deciding on item relevance and questionnaires comprehensiveness [46–48]. We used Research Electronic Data Capture (REDCap) web-based application to collect data in both Phases [49]. The electronic data collection forms were pilot tested by two children with cancer, two parents, and two HCPs before starting on data collection. Ethical approval was obtained from the Research and Ethics Board at the Hospital for Sick Children (REB# 1000064764) and the Research and Ethics Boards at the University of Toronto (Protocol# 39414) before study commencement. Written informed consents/assents were obtained from patients and parents before participation in both Phases, and from HCPs

before participation in Phase 2. Implied consents were received from HCPs and CHA researchers who participated in Phase 1.

**Phase 1: Reaching consensus on the domains and items of the WHAT questionnaires.**
*Participants*. A convenience sample of children with cancer and one of their parents was invited to participate from the Hospital for Sick Children (SickKids). Patient eligibility included children diagnosed with cancer ≥ 3 months, English-speaking, 8–18 years of age, and having used at least one type of CHA since cancer diagnosis according to self- or proxy-report. English-speaking parents of such children were recruited. Exclusion criteria for child and parent participants were severe cognitive impairments or major comorbid illnesses that could preclude questionnaires completion as determined by their treating HCPs. Oncology HCPs were recruited through the International Society of Pediatric Oncology (SIOP). General pediatric and pediatric oncology HCPs, pediatric complementary HCPs and CHA researchers were recruited through the Pediatric Complementary and Alternative Medicine (CAM) Research and Education Network (PedCAM). Eligible HCPs and researchers had practiced in their field for at least one year. We targeted the same response rate achieved in the development study of the original WHAT questionnaires (i.e., 17%) [44]; and aiming for approximately equal numbers of participants from each sample group. Delphi cycles were ceased when consensus was reached on the domains and items of the WHAT questionnaires.

*Consensus rating scale*. Participants rated the importance of each domain and item of the questionnaires using a 1 to 9 scale according to the RAND/UCLA appropriateness method (RAM) [50]. Per the RAM, a rating of 1–3 was considered "inappropriate", 4–6 considered "uncertain", and 7–9 considered "appropriate". Consensus was achieved if the median rating for a domain or an item was "appropriate" (i.e., the domain or the item had to be retained) or "inappropriate" (i.e., to be removed). Consensus was not reached if the median rating was "uncertain" or there was disagreement. Disagreement among participants occurred when 30% or more rated the appropriateness of a domain or an item from 1–3 and 30% or more rated the same domain or item from 7–9 (i.e., polarized responses) [50]. If no consensus was reached, domains and items were moved to the subsequent cycle for re-rating.

*Procedure*. REDCap electronic surveys were sent via email to children and parents. For HCPs and CHA researchers, administrative approval from SIOP and PedCAM groups was received to invite their members. Leaders of these groups sent email invitations to group members, which described the study purpose and procedure, and directed participants to the RED-Cap survey. Children were asked to rate the importance of each domain and item of the self-report version, and parents were asked to rate the proxy-report. HCPs and researchers were asked to rate both versions of the WHAT questionnaires. We also asked all participants about the appropriate terminology that children and their parents would understand without ambiguity to refer to non-prescription health products and practices. Participants were asked to choose one phrase from a list of frequently used phrases in previous research, namely: Complementary and Alternative Medicine (CAM), CHA, Traditional and Complementary Medicine (TCM), or other. Finally, participants were asked to suggest modifications in the domains and items, as well as recommendations for domains and items that might be added and were then asked to complete a demographic survey.

*Data analysis*. Descriptive statistics were used to describe participant characteristics and their ratings of domains and items. The research team reviewed narrative responses to identify any suggestions to modify or add new domains or items.

**Phase 2: Testing the face and content validity and modifying the WHAT questionnaires.** *Participants selection*. A purposive sample of children with cancer, parents, and oncology HCPs were recruited from SickKids. We also recruited parents and children online through the Ontario Parents Advocating for Children with Cancer (OPACC) group via an

email introduction by OPACC administrator. OPACC is a registered charity that is comprised of parents, guardians, survivors, and close relatives of children with cancer aiming to support children with cancer and their families across the province of Ontario, Canada. The same eligibility criteria were applied as in Phase 1. In Phase 2, we aimed to recruit for sample of children who were heterogeneous in terms of age (8–12 and 13–18 years old) and cancer diagnosis. The sample size was based on previous face and content validity testing research showing that two to three iterative cycles of cognitive interviews with a sample size of five to seven participants per cycle is sufficient [51–54].

*Procedure*. In-person or virtual individual interviews were conducted. Before the interview began, participants completed demographic and content validity ratio (CVR) forms using REDCap. The CVR form included questions that asked the participants to rate the importance of each item (i.e., relevance) using a 3-point scale (i.e., "essential", "useful but not essential", and "not necessary") [55]. During the interviews, children were asked to read over the child self-report, parents read the parent proxy-report. Children and parents were then asked to describe what they understood about the meaning of the item and whether it was easy to understand (comprehensibility), any essential items were missing (comprehensiveness), and all items and response options were relevant to the questionnaires' purpose. HCPs were asked to review both versions of the WHAT questionnaires and to answer open-ended questions about item relevance and comprehensiveness of the questionnaires. There were no questions about comprehensibility since it could only be determined by questionnaire respondents (i.e., children and parents) [46]. All individual interviews were audio-recorded and transcribed verbatim for data analysis.

*Data analysis*. At the end of each cycle, qualitative and quantitative data were triangulated to determine evidence of face and content validity and to ensure the completeness of data by combining the results from different sources [56–58]. Triangulation was conducted by analyzing and combining qualitative (interview transcripts and field notes) and quantitative (CVR and percentage agreement) data. The audio-recorded transcripts and field notes were analyzed using simple descriptive content analysis method [59]. MA & BW conducted the data analysis by reviewing each transcript and its related field notes independently and identifying the frequency of participants' concerns and perspectives about the content of the WHAT questionnaires. Disagreements were resolved by discussion. Descriptive statistics were used to describe participant characteristics. CVR was computed for each item using the following formula:

$$CVR = \frac{N_e - \frac{N}{2}}{\frac{N}{2}}$$

where, $N_e$ is the number of participants indicating "essential" and N is the total number of participants [55]. If more than half of the participants rated an item as essential (i.e., positive CVR), the item was considered relevant [55]. Percent agreement was calculated to determine rating agreements between two groups: children and HCPs for the child version, and parents and HCP for the parent version. Good agreement was defined as at least 75% of participants rating an item as essential or useful [44]. Final decisions on modifying, removing, or retaining items in the modified WHAT were made by considering all findings from different data sources [44, 46].

## Results

### Phase 1: Reaching consensus on the domains and items of the WHAT questionnaires

**Participants.** The Delphi survey was completed by 61 participants with response rate per group as 68% (15/22) of children, 59% (13/22) of parents, 24% (10/41) of PedCAM pediatric HCPs and complementary HCPs and researchers, and 30% (23/77) of SIOP oncology HCPs. Table 1 shows the characteristics of the Delphi study participants.

**Delphi survey results.** A total of 80.3% (49/61) of participants thought that the appropriate terms to refer to health products and practices that exists outside mainstream healthcare were CAM (28/49, 57%) or CHA (21/49, 43%). Only 16.4% (10/61) chose the term TCM, and 3.3% (2/61) suggested other terms (i.e., Holistic health modalities, over the counter, allied health partners). All domains and items were agreed upon by all participants and consensus was reached after one cycle of Delphi Survey (see Table 2).

### Phase 2: Testing face and content validity and modifying the WHAT questionnaires

**Participants.** Forty participants (12 children, 14 parents, 14 HCPs) completed two iterative cycles of individual interviews. The sample of children was heterogeneous in terms of age groups and cancer diagnosis. HCPs were experienced in the field with various professional backgrounds. Table 3 shows the characteristics of participants.

**First cycle of the face and content validity testing.** Five children, seven parents, and seven HCPs, who did not participate in Phase 1, completed the individual interviews. The interview duration was 30.9 minutes on average (SD = 7.5). Table 4 shows the findings of the first cycle of individual interviews. The participants confirmed that the WHAT questionnaires included comprehensive lists of questions. However, they did not find the paper format and the use of tables in the WHAT as user-friendly—limited space and a busy layout. A discussion within the research team concluded to redesign the WHAT as electronic versions, and to remove all tables and replace them with multiple choice and open-ended questions. Moreover, all items of the original WHAT questionnaires were meant to be generic. Many participants felt that the generic language of the questionnaires was vague and without time bound for many items. To minimize the ambiguity, participants suggested rephrasing the questions to be cancer-relevant and linking the timeframe to cancer events, like cancer diagnosis. Based on this, the research team decided to edit all items to be cancer specific. For item relevance, participants felt that the two versions of the WHAT included relevant questions about CHA use. However, a few items were not felt to be essential by some participants according to CVR scores (see Table 5). During the interviews, some HCPs stated that item five asked about information that could be "good to know" but not clinically essential. No other reasons for negative CVR were captured during the interviews from children and parents. The research team reviewed transcripts and CVR, and concluded that the generic nature of the questions might be the reason for the negative CVR scores. Thus, the research team decided to keep all items of both WHAT versions, with revision to be cancer-specific according to the comprehensibility testing results, and then verify their relevance with a new group of children, parents, and HCPs in the second cycle.

**Resulting modified WHAT questionnaires.** Based on the results of the first cycle of the face and content validity testing, the research team modified the items of the original WHAT questionnaires and designed electronic versions using REDCap, which would ideally be integrated into the electronic health record application for use clinically. The modified self- (13

**Table 1. Characteristics of participants who participated in the Delphi survey.**

| | Children with Cancer (n = 15) | Parents (n = 13) |
|---|---|---|
| Age, mean (SD) | 14.04 (3.1) | 44.7 (5.2) |
| Sex, female n (%) | 6 (40%) | 12 (92%) |
| Educational Background, n (%) | | |
| Grade 3–7 (8–12 years) | 5 (33.3%) | — |
| Grade 8–12 (13–18 years) | 10 (66.7%) | — |
| Graduated Secondary School | — | 1 (7.7%) |
| Some College/Technical School | — | 1 (7.7%) |
| Graduated College/Technical School | — | 4 (30.7%) |
| Some Graduate School | — | 1 (7.7%) |
| Graduate Degree | — | 6 (46.2%) |
| Child's Diagnosis, n (%) | | |
| Acute Lymphoblastic Leukemia | 4 (27%) | 4 (30.7%) |
| Acute Myeloid Leukemia | 2 (13%) | 0 |
| Brain Tumor | 3 (20%) | 3 (23.1%) |
| Neuroblastoma | 1 (7%) | 2 (15.4%) |
| Osteosarcoma | 1 (7%) | 0 |
| Ewing's Sarcoma | 2 (13%) | 2 (15.4%) |
| Other | 2 (13%) | 2 (15.4%) |
| Child's disease duration in years, median (range) | 1.25 (0.33–11.42) | 1.33 (0.25–11.33) |
| Ethnicity, n (%) | | |
| South Asian | 1 (6.7%) | 2 (15%) |
| South East Asian | 1 (6.7%) | 1 (7.7%) |
| White (Caucasian) | 7 (46.6%) | 7 (53.8%) |
| Other | 3 (20%) | 2 (15.4%) |
| Prefer not to answer | 1 (6.7%) | 1 (7.7%) |
| Missing data | 2 (13.3%) | 0 |
| Country, n (%) | | |
| Canada | 15 (100%) | 13 (100%) |
| | **Oncology HCPs (n = 21)** | **Complementary HCPs and researchers (n = 12)** |
| Age, mean (SD) | 44.3 (12.3) | 47.2 (12.1) |
| Sex, female n (%) | 14 (67%) | 5 (34%) |
| Profession, n (%) | | |
| Pediatric Oncologist | 8 (38.1%) | — |
| Pediatric Oncology Nurse | 7 (33.3%) | — |
| Oncology Nurse Practitioner | 3 (14.3%) | — |
| Other HCPs | 3 (14.3%) | — |
| Naturopathic Practitioner | — | 5 (41.6%) |
| Chiropractor | — | 2 (16.7%) |
| Massage Therapist | — | 2 (16.7%) |
| Other CHA HCPs and Researcher | — | 3 (25%) |
| Years of experience, mean (SD) | 19.67 (12.17) | 19.08 (11.94) |
| Country, n (%) | | |
| United States | 7 (33%) | 0 |
| Canada | 4 (19%) | 10 (83%) |
| Kenya | 3 (14%) | 0 |
| Turkey | 2 (10%) | 0 |
| Cameroon | 1 (5%) | 0 |

*(Continued)*

**Table 1.** (Continued)

| Chile | 1 (5%) | 0 |
|---|---|---|
| India | 1 (5%) | 0 |
| Netherlands | 1 (5%) | 0 |
| Uruguay | 1 (5%) | 0 |
| Jordan | 0 | 1 (8%) |
| China | 0 | 1 (8%) |

items, S2 Appendix) and proxy-report (15 items, S3 Appendix) versions of the WHAT were cancer-specific and asked about the use of CHA based on three timeframes: since cancer diagnosis, in the past four weeks, and future use [49]. The research team also added examples to make the questions easier to complete. CHA terminology, definition, and classification from the US National Center for Complementary and Integrative Health (NCCIH) were adapted, which were the most cited in the past research [1, 3]; and using the term "CHA" was consistent with our Delphi findings.

**Second cycle of the face and content validity testing.** A new sample of seven children, seven parents, and seven HCPs completed the individual interviews. The same procedure as the first cycle was followed. The interview duration was 25.4 minutes on average (SD = 7.7). Table 6 shows the findings of the second cycle of individual interviews. All participants confirmed that the questionnaires included comprehensive short lists of questions for clinical use. All children and parents understood all items of the modified WHAT questionnaires except two children (8 and 9 years old) who could not independently read, understand, or answer all questions without help from parents or the researcher. So, the research team changed the targeted age group for the self-report version and let parents help young children (i.e., 8 to 11 years) complete the child version. The two versions of the modified WHAT questionnaires were determined by all participants to represent relevant questions on CHA use in pediatric oncology (see Table 5 for CVR).

## Discussion

This article presents the results of adapting and testing the face and content validity of the WHAT questionnaires for multidimensional clinical assessment of CHA use in pediatric oncology. We introduce the first electronic disease-specific CHA questionnaires (i.e., the modified WHAT) that have evidence of face and content validity and are designed to assess CHA use and initiate clinical discussions about CHA in pediatric oncology. The lack of such CHA questionnaires was considered a gap as reported in previous systematic reviews [3, 6, 7]. Also, previous research has supported the need for open dialogue in pediatric oncology settings about CHA and the resultant need to assess the use of CHA using a questionnaire [17–19, 33, 44]. Open and non-judgemental discussion about CHA use in clinical settings would be a key step to enhance disclosure rates, minimize the potential risk of interactions with conventional treatments, and enhance HCPs understanding of CHA use and its impact on the child's health. Ultimately, adapted and tailored assessment questionnaires will enrich HCPs knowledge about CHA, identify safe and effective CHA products and practices, and may pave the way toward integrative pediatric oncology care.

Results from the Delphi survey showed that most participants (80%) were familiar with the terms CAM and CHA. Accordingly, we used the term CHA in the modified WHAT questionnaires to refer to the products and practices that children with cancer may use on their own or

**Table 2. Median rating of each domain and items of the two versions of the WHAT questionnaires after one cycle of the Delphi survey.**

| The Youth Version of the WHAT Questionnaire | Children with Cancer (n = 15) | | | Complementary HCPs and CHA Researchers (n = 12) | | | HCPs (n = 21) | | | Percentage Agreement[c] | Final Decision |
|---|---|---|---|---|---|---|---|---|---|---|---|
| | Median | % ≥ 7[a] | % ≤ 3[b] | Median | % ≥ 7[a] | % ≤ 3[b] | Median | % ≥7[a] | % ≤ 3[b] | | |
| **Domains** | | | | | | | | | | | |
| Child's use of CHA (items 2, 3, 7, 7a, 8) | 8 | 93% | 0% | 8 | 83% | 8% | 9 | 100% | 0% | 94% | Include domain |
| Factors associated with CHA use (items 5, 6, 7b) | 8 | 80% | 0% | 8 | 92% | 0% | 9 | 90% | 0% | 88% | Include domain |
| Perceived impact of CHA use (items 7c, 7d, 7e) | 8 | 93% | 0% | 8 | 83% | 8% | 9 | 100% | 0% | 94% | Include domain |
| Communication about CHA (items 1, 4) | 8 | 87% | 0% | 8 | 52% | 0% | 9 | 95% | 0% | 92% | Include domain |
| **Sections and Items** | | | | | | | | | | | |
| Section 1: Conventional treatments | 8 | 87% | 7% | 8.5 | 75% | 8% | 8 | 86% | 0% | 83% | Include section |
| Section 2: CHA definition and examples | 8 | 93% | 0% | 7.5 | 75% | 8% | 9 | 95% | 0% | 90% | Include section |
| Item 1: Discussing CHA with health care team | 8 | 100% | 0% | 8.5 | 48% | 8% | 9 | 95% | 0% | 94% | Include item |
| Item 2: Past use of CHA | 8 | 93% | 0% | 8 | 67% | 25% | 8 | 90% | 0% | 85% | Include item |
| Item 3: Types of CHA used in the past | 8 | 87% | 0% | 7.5 | 58% | 17% | 9 | 95% | 0% | 83% | Include item |
| Item 4: Discussing CHA with family | 8 | 93% | 0% | 7.5 | 38% | 8% | 8 | 76% | 0% | 79% | Include item |
| Item 5: Person who decided to use CHA | 8 | 93% | 0% | 7.5 | 58% | 17% | 8 | 81% | 0% | 79% | Include item |
| Item 6: Change on medical treatments because of CHA | 8 | 80% | 0% | 8.5 | 92% | 8% | 9 | 90% | 0% | 88% | Include item |
| Item 7: Recent use of CHA (past two weeks) | 8 | 87% | 0% | 8 | 75% | 17% | 9 | 86% | 5% | 83% | Include item |
| Item 7a: Types of CHA used recently | 8 | 93% | 0% | 7.5 | 58% | 17% | 9 | 86% | 5% | 81% | Include item |
| Item 7b: Reasons of recent use of CHA | 8 | 100% | 0% | 8 | 75% | 0% | 9 | 86% | 5% | 88% | Include item |
| Item 7c: How helpful was the recent use | 8 | 100% | 0% | 8 | 83% | 8% | 8 | 81% | 0% | 88% | Include item |
| Item 7d: Benefits of recent use of CHA | 8 | 93% | 0% | 8 | 75% | 0% | 8 | 76% | 0% | 81% | Include item |
| Item 7e: Risks/cons of recent use of CHA | 8 | 87% | 0% | 8 | 67% | 0% | 8 | 81% | 0% | 79% | Include item |
| Item 8: Future use of CHA | 8 | 80% | 0% | 8 | 83% | 8% | 8 | 81% | 0% | 81% | Include item |
| **The Parent Version of the WHAT Questionnaire** | **Parents (n = 13)** | | | **Complementary HCPs and CHA Researchers (n = 12)** | | | **HCPs (n = 21)** | | | **Percentage Agreement[c]** | **Final Decision** |
| | Median | % ≥ 7[a] | % ≤ 3[b] | Median | % ≥ 7[a] | % ≤ 3[b] | Median | % ≥7[a] | % ≤ 3[b] | | |
| **Domains** | | | | | | | | | | | |
| Child's use of CHA (items 3, 4, 8, 8a, 9) | 8 | 92% | 0% | 8 | 83% | 8% | 9 | 100% | 0% | 93% | Include domain |
| Factors associated with use of CHA (items 2, 6, 7, 8c) | 8 | 92% | 0% | 8 | 92% | 0% | 9 | 90% | 0% | 91% | Include domain |
| Perceived impact of CHA use (items 8d, 8e, 8f) | 8 | 85% | 0% | 8 | 83% | 8% | 9 | 100% | 0% | 91% | Include domain |
| Communication about CHA (items 1, 5, 8b) | 8 | 92% | 8% | 8 | 52% | 0% | 9 | 95% | 0% | 93% | Include domain |
| **Sections and items** | | | | | | | | | | | |
| Section 1: Conventional treatments | 9 | 85% | 0% | 9 | 83% | 8% | 8 | 90% | 0% | 87% | Include section |
| Section 2: CHA definition and examples | 8 | 92% | 0% | 8 | 83% | 8% | 9 | 100% | 0% | 93% | Include section |
| Item 1: Discussing CHA with child's HCP | 9 | 92% | 0% | 8.5 | 48% | 8% | 9 | 95% | 0% | 91% | Include item |
| Item 2: Troubles getting CHA (i.e., access) | 8 | 85% | 8% | 7 | 58% | 0% | 9 | 76% | 10% | 76% | Include item |

*(Continued)*

**Table 2.** (Continued)

| | | | | | | | | | | |
|---|---|---|---|---|---|---|---|---|---|---|
| Item 3: Child's past use of CHA | 8 | 85% | 0% | 8 | 67% | 25% | 9 | 100% | 0% | 87% | Include item |
| Item 4: Types of CHA used in the past by the child | 9 | 85% | 0% | 7 | 58% | 17% | 9 | 95% | 0% | 83% | Include item |
| Item 5: Discussing CHA with the child | 8 | 77% | 0% | 7.5 | 38% | 8% | 8 | 81% | 0% | 76% | Include item |
| Item 6: Person who decided on child's use of CHA | 8 | 85% | 0% | 7.5 | 58% | 17% | 8 | 81% | 0% | 89% | Include item |
| Item 7: Change on medical treatments because of CHA | 9 | 85% | 0% | 8.5 | 92% | 8% | 9 | 90% | 0% | 85% | Include item |
| Item 8: Child's recent use of CHA (past 2 weeks) | 8 | 92% | 0% | 8 | 75% | 8% | 9 | 90% | 10% | 87% | Include item |
| Item 8a: Types of CHA used recently by the child | 8 | 92% | 0% | 7.5 | 67% | 8% | 9 | 90% | 10% | 85% | Include item |
| Item 8b: Consult someone for child's use of CHA | 8 | 77% | 0% | 7.5 | 48% | 0% | 9 | 90% | 0% | 85% | Include item |
| Item 8c: Reasons of recent use of CHA by child | 8 | 100% | 0% | 8 | 75% | 0% | 9 | 90% | 5% | 78% | Include item |
| Item 8d: How helpful was the child's recent use | 8 | 92% | 0% | 8 | 92% | 8% | 9 | 95% | 0% | 93% | Include item |
| Item 8e: Benefits of recent use of CHA by child | 8 | 92% | 0% | 8 | 100% | 0% | 9 | 95% | 0% | 96% | Include item |
| Item 8f: Risks/cons of the child's recent use | 8 | 100% | 0% | 8 | 92% | 0% | 9 | 95% | 0% | 96% | Include item |
| Item 9: Future use of CHA by the child | 9 | 92% | 0% | 8 | 83% | 8% | 8 | 81% | 0% | 85% | Include item |

[a] Percentage of participants who rated the domain or the item as more than or equal 7 on the scale from 1 to 9 (RAM method)

[b] Percentage of participants who rated the domain or the item as less than or equal to 3 on the scale from 1 to 9 (RAM method)

[c] Percentage of participants from the three groups who rated the domain or the item as more than or equal to 7

be recommended by their parents or caregivers without medical prescription. We also included a section to define CHA with examples of the frequently used CHA products and practices by children with cancer. Including friendly terminology and a clear definition of CHA should help respondents understand the subsequent questions about CHA, and ultimately minimize cognitive burden associated with questionnaire completion and response errors. Also, we retained the section about conventional treatments, simplified it, and modified its wording to ask about the prescribed therapies by primary health care team for cancer care (i.e., oppose to the generic format), so respondents would actively distinguish between CHA and conventional cancer treatments.

We engaged children, parents, and HCPs in refining and testing the face and content validity of the modified WHAT questionnaires. This has not been the case for all existing pediatric oncology CHA questionnaires [3, 8]. Engaging children and parents in designing patient-reported outcome questionnaires is essential to capture their perspectives on whether the included items are relevant to the construct of interest and in the context of use. Also, the child's and the parent's perspectives are vital to determine whether the questionnaire is comprehensive and easily understood by respondents [46, 61, 62]. Moreover, using a sequential phased approach ensured iterative adjustments of the WHAT questionnaires and testing the face and content validity by children, parents, and HCPs ensured the inclusion of essential domains and items that are consistent with the measurement purpose of the modified WHAT questionnaires.

**Table 3. Characteristics of participants in the face and content validity study.**

| Children with Cancer | Cycle 1 (n = 5) | Cycle 2 (n = 7) |
|---|---|---|
| Age, mean (SD) | 13.45 (2.96) | 12.56 (2.65) |
| Sex, female, n (%) | 3 (60%) | 4 (57%) |
| Educational Background, n (%) | | |
| Grade 3–7 | 2 (40%) | 3 (43%) |
| Grade 8–12 | 3 (60%) | 4 (57%) |
| Child's Diagnosis, n (%) | | |
| Acute Lymphoblastic Leukemia | 1 (20%) | 2 (29%) |
| Acute Myeloid Leukemia | 1 (20%) | 0 |
| Lymphoma | 1 (20%) | 1 (14%) |
| Neuroblastoma | 0 | 1 (14%) |
| Osteosarcoma | 0 | 2 (29%) |
| Ewing's Sarcoma | 2 (40%) | 0 |
| Other | 0 | 1 (14%) |
| Mean disease duration in years, median (range) | 4 (0.42–6.50) | 0.75 (0.25–4.75) |
| Ethnicity, n (%) | | |
| Aboriginal (Inuit, Metis, North American Indian) | 1 (20%) | 0 |
| Arab/West Asian | 2 (40%) | 1 (14%) |
| South Asian | 1 (20%) | 1 (14%) |
| White (Caucasian) | 2 (40%) | 3 (43%) |
| Other | 0 | 2 (29%) |
| Prefer not to answer | 1 (20%) | 0 |
| **Parents of Children with Cancer** | **Cycle 1** (n = 7) | **Cycle 2** (n = 7) |
| Age, mean (SD) | 43.14 (6.66) | 43.98 (5.28) |
| Sex, female n (%) | 6 (86%) | 6 (86%) |
| Educational Background, n (%) | | |
| Graduated Secondary School | 1 (14.3%) | 0 |
| Some College/Technical School | 0 | 1 (14.3%) |
| Graduated College/Technical School | 1 (14.3%) | 2 (28.6%) |
| Graduate Degree | 5 (71.4%) | 4 (57.1%) |
| Income, n (%) | | |
| Less than $24,999 | 1 (14.3%) | 1 (14.3%) |
| $25,000 to $49,999 | 2 (28.6%) | 0 |
| $75,000 to $99,999 | 1 (14.3%) | 1 (14.3%) |
| $100,000 to $149,999 | 0 | 1 (14.3%) |
| $200,000 or more | 0 | 1 (14.3%) |
| Prefer not to answer | 3 (42.8%) | 3 (42.8%) |
| Parents' expectations of using CHA for their children, n (%) | | |
| Using CHA would make my child feel better | 3 (43%) | 6 (86%) |
| Using CHA would prevent symptom of my child | 2 (29%) | 6 (86%) |
| Using CHA would cure my child's illness | 2 (29%) | 2 (29%) |
| Using CHA would treat my child's symptoms | 2 (29%) | 5 (71%) |
| Using CHA is safe | 2 (29%) | 7 (100%) |
| Other | 1 (14%) | 0 |
| Parent's use of CHA, Yes, n (%) | 5 (71%) | 4 (57%) |
| Child's Diagnosis, n (%) | | |
| Acute Lymphoblastic Leukemia | 3 (42.8%) | 1 (14.3%) |
| Acute Myeloid Leukemia | 1 (14.3%) | 0 |

(*Continued*)

**Table 3.** (Continued)

| | | |
|---|---|---|
| Lymphoma | 1 (14.3%) | 1 (14.3%) |
| Brain Tumor | 0 | 1 (14.3%) |
| Neuroblastoma | 0 | 1 (14.3%) |
| Osteosarcoma | 0 | 2 (28.5%) |
| Ewing's Sarcoma | 2 (28.6%) | 0 |
| Other | 0 | 1 (14.3%) |
| Child's disease duration in years, median (range) | 4 (042–11.42) | 0.58 (0.25–4.75) |
| Ethnicity, n (%) | | |
| Aboriginal (Inuit, Metis, North American Indian) | 1 (14.3%) | 0 |
| Arab/West Asian | 2 (28.5%) | 1 (14.3%) |
| South Asian | 1 (14.3%) | 1 (14.3%) |
| White (Caucasian) | 1 (14.3%) | 4 (57.1%) |
| Other | 1 (14.3%) | 1 (14.3%) |
| Prefer not to answer | 1 (14.3%) | 0 |
| **Oncology HCPs** | **Cycle 1** (n = 7) | **Cycle 2** (n = 7) |
| Age, mean (SD) | 52.29 (13.34) | 40.14 (10.33) |
| Sex, Female n (%) | 5 (71%) | 7 (100%) |
| Profession, n (%) | | |
| Pediatric oncologist | 3 (42.8%) | 1 (14.3%) |
| Pediatric oncology nurse practitioner | 2 (28.6%) | 1 (14.3%) |
| Master-prepared pediatric oncology nurse | 0 | 3 (42.8%) |
| Pediatric oncology pharmacist | 0 | 1 (14.3%) |
| Pediatric oncology fellow | 2 (28.6%) | 1 (14.3%) |
| Years of experience, mean (SD) | 25.29 (12.51) | 13.64 (9.39) |
| Discuss CHA with patients/parents, n (%) | | |
| Never | 0 | 0 |
| Rarely | 2 (28.6%) | 4 (57.1%) |
| Sometimes | 1 (14.3%) | 1 (14.3%) |
| Most of the time | 4 (57.1%) | 0 |
| Always | 0 | 2 (28.6%) |
| Provide advice to patients/parents about CHA, n (%) | | |
| Never | 0 | 0 |
| Rarely | 2 (28.6%) | 5 (71.4%) |
| Sometimes | 3 (42.8%) | 1 (14.3%) |
| Most of the time | 2 (28.6%) | 0 |
| Always | 0 | 1 (14.3%) |

In this research project, we adapted pediatric generic short CHA questionnaires and used their contents as the core sets of domains and items in the Delphi survey. We also modified them into electronic disease-specific questionnaires in the face and content validity study. This is consistent with a previously recommended method of developing and validating an adult oncology CHA questionnaire [63]. Researchers suggested designing a standardized CHA questionnaire to assess CHA use across adult populations, which would help researchers generate a meaningful comparison among different populations by using a core set of questions (i.e., generic) that could be adapted and refined later as disease-specific questionnaires, like for cancer and arthritis [63]. The results of our first cycle of the face and content validity testing revealed that the generic items of the questionnaires were vague and not meaningful for

**Table 4. Findings of the first cycle of the individual interviews regarding the comprehensiveness, comprehensibility, and relevance.**

| | Children n = 5 | HCPs n = 7 | Parents n = 7 | Final Decision by the Research Team |
|---|---|---|---|---|
| Comprehensiveness | • The child version was thought to be comprehensive.<br>• No extra items were suggested. | • Both versions were thought to be comprehensive.<br>• Extra items were suggested by HCPs (each one came from a different HCP) to ask about: cannabis use, the difference between complementary and alternative approaches, parents' use of CHA, and whether HCPs discuss CHA with children and parents. | • The parent version was thought to be comprehensive.<br>• One parent suggested adding one item to ask about the cultural background. | • Adding cannabis as an example of CHA to both versions of the WHAT is consistent with the purpose of the questionnaires because there is a trend of using cannabis by children with cancer [60], with limited information about its safety and efficacy.<br>• Other suggested items are irrelevant to the purpose of the WHAT as short clinical questionnaires, and they can be added to the WHAT instruction manual (i.e., instructions on how to use the WHAT questionnaires in clinical practice) as potential questions that HCPs would ask during discussions about CHA with children and parents. |
| Comprehensibility | • The paper format and the use of tables in the WHAT were not user-friendly; for example, most children found section 1 was vague in terms of the timeline (past or current) and type of treatments (cancer-directed or other), and challenging to answer, given the large number of conventional therapies that had to be written in a small space in the table. | • Most HCPs supported the idea that tables in both versions would be burdensome, and respondents might likely skip them. | • The paper format and the use of tables in the WHAT were not user-friendly; for example, most parents found section 1 confusing about whether to include conventional therapies that children were on at home or receiving in the hospital and mentioned the concern about how difficult to answer this section was, given the large number of conventional therapies that had to be written in a small space in the table. | • Redesign both versions of the WHAT as electronic versions.<br>• Remove all tables from both WHAT versions and replace them with multiple-choice and open-ended questions.<br>• Skip logic branching had also been applied, i.e., showing or hiding the follow-up questions based on answers to an earlier question, which would help minimize the respondents' cognitive burden. |
| | • The generic language of the questionnaires was vague and without time bound for many items. | • Asking cancer-specific questions would help HCPs understand the complexities of CHA use in cancer. | • The generic language of the questionnaires was vague and without time bound for many items. | • Edit all items of both WHAT versions to be cancer specific. |
| | • The purpose of the child version questionnaire was clear, except for two younger children, both 10 years of age.<br>• Most children were unfamiliar with the word "conventional," and the two younger children could not understand most questions without help from their parents or researchers. | • The purpose of both versions of the questionnaires was clear. | • The purpose of the parent version questionnaire was clear.<br>• Most parents recommended modifying the word "conventional" with a word frequently used in clinical settings, like the primary health care team for cancer care. | • Replace the vague words according to the participants' suggestions.<br>• Test the comprehensibility of the child version in the next cycle with a sample of 2/7 (28.6%) young children (i.e., 8–11 years) to confirm the age group of the modified WHAT questionnaires. |
| Relevance | • All items of the self-report were relevant.<br>• No explanation for the negative CVR (see Table 5 for CVR ratings). | • All items were relevant (both versions).<br>• Item five asked about information that could be "good to know," (see also Table 5 for CVR). | • All items of the proxy-report were relevant.<br>• No explanation for the negative CVR (see Table 5 for CVR ratings). | • Keep all items of both WHAT versions—according to data triangulation—with revision to be cancer specific and verify their relevance with a new group of children, parents, and HCPs in the second cycle. |

**Table 5. Agreement by participants regarding the relevance of each item of the WHAT questionnaires.**

| The Youth Version of the Modified WHAT Questionnaire | Cycle 1 | | | | Cycle 2 | | | | Final Decision |
|---|---|---|---|---|---|---|---|---|---|
| | Children n = 5 | HCPs n = 7 | Total n = 12 | Percentage Agreement | Children n = 7 | HCPs n = 7 | Total n = 14 | Percentage Agreement | |
| | CVR[a] | CVR[a] | n[b] | (%)[b] | CVR[a] | CVR[a] | n[b] | (%)[b] | |
| Section 1: Conventional treatments | 0.6 | 0.71 | 11 | 92% | 0.71 | 1.00 | 14 | 100% | Include section |
| Section 2: CHA definition and examples | 0.6 | 1.00 | 12 | 100% | -0.14 | 1.00 | 14 | 100% | Include section |
| Item 1: CHA use since cancer diagnosis | 0.2 | 0.71 | 12 | 100% | 0.14 | 1.00 | 14 | 100% | Include item |
| Item 2: Used types of CHA by the child since cancer diagnosis | -0.6 | 1.00 | 12 | 100% | 0.71 | 1.00 | 14 | 100% | Include item |
| Item 3: Discussion between Patients and HCP about CHA use | -0.2 | 0.43 | 12 | 100% | 0.14 | 0.71 | 14 | 100% | Include item |
| Item 4: Who decided that the child should use CHA | -0.2 | 0.71 | 12 | 100% | 0.43 | 0.71 | 14 | 100% | Include item |
| Item 5: Discussion between patients and parents about CHA use | -1 | -0.43 | 11 | 92% | -0.14 | 0.43 | 13 | 93% | Include item |
| Item 6: Have the child changed cancer treatment for CHA use | 0.2 | 0.71 | 12 | 100% | 0.14 | 0.71 | 13 | 93% | Include item |
| Item 7: CHA use during the past four weeks (recent use) | -1 | 0.43 | 12 | 100% | 0.14 | 0.71 | 14 | 100% | Include item |
| Item 7a: Used types of CHA by the child in the past four weeks | 0.2 | 0.71 | 12 | 100% | 0.43 | 0.43 | 14 | 100% | Include item |
| Item 7b: Reasons of recent use of CHA by the child | 0.2 | -0.14 | 12 | 100% | 0.43 | 0.71 | 14 | 100% | Include item |
| Item 7c: How helpful was the recent use of CHA | -0.2 | 1.00 | 12 | 100% | 0.43 | 0.71 | 14 | 100% | Include item |
| Item 7d: What were the benefit of recent use of CHA | 0.6 | 1.00 | 12 | 100% | 0.43 | 0.71 | 13 | 93% | Include item |
| Item 7e: What were the risks of the recent use of CHA | 0.6 | 1.00 | 12 | 100% | 0.71 | 1.00 | 13 | 93% | Include item |
| Item 8: Plans for future use of CHA by the child | -0.2 | 0.43 | 11 | 92% | 0.43 | 0.71 | 14 | 100% | Include item |
| **The Parent Version of the Modified WHAT Questionnaire** | **Cycle 1** | | | | **Cycle 2** | | | | Final Decision |
| | Parents n = 7 | HCPs n = 7 | Total n = 14 | Percentage Agreement | Parents n = 7 | HCPs n = 7 | Total n = 14 | Percentage Agreement | |
| | CVR[a] | CVR[a] | n[b] | (%)[b] | CVR[a] | CVR[a] | n[b] | (%)[b] | |
| Section 1: Conventional Cancer treatments | 0.71 | 0.71 | 13 | 93% | 1.00 | 0.71 | 14 | 100% | Include section |
| Section 2: CHA definition and examples of common CHA used in Pediatric Oncology | 0.43 | 1.00 | 14 | 100% | 0.43 | 0.71 | 14 | 100% | Include section |
| Item 1: Child's CHA use since cancer diagnosis | 0.71 | 1.00 | 14 | 100% | 0.14 | 0.71 | 14 | 100% | Include item |
| Item 2: Used types of CHA by the child since cancer diagnosis | 0.71 | 1.00 | 14 | 100% | 0.71 | 0.71 | 14 | 100% | Include item |
| Item 3: Discussion between Parents and HCP about CHA use | 0.43 | 0.71 | 14 | 100% | 0.43 | 0.71 | 14 | 100% | Include item |
| Item 4: Who decided that the child should use CHA | 0.43 | 0.71 | 14 | 100% | 0.14 | 0.71 | 14 | 100% | Include item |
| Item 5: Discussion between patients and parents about CHA use | 0.14 | -0.71 | 13 | 93% | 0.43 | 0.14 | 14 | 100% | Include item |
| Item 6: Have Parents changed cancer treatment for CHA use | 0.71 | 0.71 | 14 | 100% | 0.43 | 0.43 | 14 | 100% | Include item |

*(Continued)*

**Table 5.** (Continued)

| | | | | | | | | | |
|---|---|---|---|---|---|---|---|---|---|
| Item 7: Difficulty accessing CHA | 0.43 | 0.14 | 14 | 100% | 0.14 | 0.14 | 13 | 93% | Include item |
| Item 8: Child's CHA use during the past four weeks (recent use) | 0.71 | 0.71 | 14 | 100% | 0.43 | 0.43 | 14 | 100% | Include item |
| Item 8a: Used types of CHA by the child in the past four weeks | 0.71 | 0.71 | 13 | 93% | 0.71 | 0.43 | 14 | 100% | Include item |
| Item 8b: Did Parents consult someone to use CHA for the child | 0.43 | 0.71 | 14 | 100% | -0.14 | 0.43 | 13 | 93% | Include item |
| Item 8c: Reasons of recent use of CHA by the child | 0.43 | 0.43 | 14 | 100% | 0.14 | 0.43 | 13 | 93% | Include item |
| Item 8d: How helpful was the recent use of CHA | 1.00 | 0.71 | 14 | 100% | 0.14 | 0.71 | 14 | 100% | Include item |
| Item 8e: What were the benefit of recent use of CHA | 1.00 | 0.71 | 14 | 100% | 0.71 | 0.71 | 13 | 93% | Include item |
| Item 8f: What were the risks of the recent use of CHA | 1.00 | 0.71 | 14 | 100% | 0.43 | 1.00 | 13 | 93% | Include item |
| Item 9: Plans for future use of CHA by the child | 0.71 | 0.43 | 14 | 100% | 0.43 | 0.71 | 14 | 100% | Include item |

CVR ratings of generic items and modified cancer-specific items of the questionnaires are presented under cycle 1 and cycle 2 columns, respectively.

[a]CVR > 0 is considered positive CVR which means more than half of the sample rated the item as "essential"

[b]Number and percentage of participants who found the item to be essential or useful. Good percentage agreement being defined as at least 75% rated an item as "essential" or "useful"

children with cancer and their parents. HCPs highlighted that asking cancer-specific questions would help HCPs understand the complexities of CHA use in cancer, as opposed to the general use of CHA. Therefore, we refined all items to be cancer-specific to enhance the comprehensibility and offer questionnaires that would help HCPs describe CHA use in pediatric oncology. Also, electronic questionnaires versions were found to be more user-friendly and flexible, especially for some complex questions. Although the main purpose of the modified WHAT questionnaires is to facilitate HCPs discussion about CHA with children and parents, the collected data could also be used for research purposes and enable standardized comparison between pediatric oncology findings and other studies in pediatric conditions using the original generic WHAT versions.

Most of the existing pediatric oncology CHA questionnaires were designed to be used as parent-proxy report [3, 8]. We have now generated face and content validity tested self- and proxy-report CHA questionnaires for use in clinical pediatric oncology settings. A self-report questionnaire will support direct understanding of child's CHA use from the child and can facilitate their active participation in discussions with HCPs. It can be argued that parent proxy-report questionnaires can be acceptable, especially as parents are routinely asked by HCPs to comment on their child's health status. Agreement between the self- and proxy-report is not granted unless confirmed by inter-rater reliability testing; for example, previous research has shown that proxy-report results are not equivalent to the self-report in a study on perceptions of quality of life in children under chiropractic care [64]. Therefore, the two versions of the modified WHAT could be used to compare the responses on self- and proxy-reports about CHA use in pediatric oncology, which was not explored previously.

Comparable to the original development study of the WHAT questionnaires, we used the COSMIN checklist to guide refining and initially validating the two versions of the WHAT questionnaires [44, 46]. This was used to ensure a rigorous testing process, which was not the

**Table 6. Findings of the second cycle of the individual interviews regarding the comprehensiveness, comprehensibility, and relevance.**

| | Children n = 7 | HCPs n = 7 | Parents n = 7 | Final Decision by the Research Team |
|---|---|---|---|---|
| Comprehensiveness | • The child version was thought to be comprehensive and included short lists of questions for clinical use and would be sufficient to describe the use of CHA by children with cancer.<br>• No extra items were suggested. | • Both versions were thought to be comprehensive and included short lists of questions for clinical use and would be sufficient to describe the use of CHA by children with cancer.<br>• No extra items were suggested | • The parent version was thought to be comprehensive and included short lists of questions for clinical use and would be sufficient to describe the use of CHA by children with cancer.<br>• One parent suggested adding one item to ask about CHA frequency, but the same parent mentioned that it would add an extra burden on the respondents. | • The research team discussed the suggestion by the parent and decided that this item would be difficult to measure (i.e., too precise to be measured in a brief clinical questionnaire) and it could be added to the instruction manual as a potential point of discussion by HCPs with children and their parents during clinic visits. |
| Comprehensibility | • The purpose of the child version questionnaire was clear, except for two younger children, 8 and 9 years old.<br>• All children understood all items of the child version of the modified WHAT questionnaires except the two younger children.<br>• No suggestions were received to revise any item. | • The purpose of both versions of the questionnaires was clear.<br>• No suggestions were received to revise any item. | • The purpose of the parent version questionnaire was clear.<br>• All parents understood all items of the parent version of the modified WHAT questionnaires.<br>• No suggestions were received to revise any item. | • For the child self-report version, parents or caregivers can help young children (8 to 11 years) complete the questionnaire.<br>• Older children (12 to 18 years) would complete the self-report independently.<br>• No need for further changes. |
| Relevance | • All items were relevant.<br>• No explanation for the negative CVRs (see Table 5 for CVR ratings). | • All items in both versions were relevant (see Table 5 for CVR ratings). | • All items were relevant.<br>• For item 8b (negative CVR ratings, see Table 5), during the interviews, most parents felt that this item would be important since this would reflect whether the recent use of CHA was based on recommendations from a reliable source. | • Keep all modified items of both versions of the WHAT questionnaires according to data triangulation.<br>• No need for further changes. |

case with most CHA questionnaires across pediatric populations [3, 8]. Contrary to most existing CHA questionnaires, we also identified the measurement purpose of the modified WHAT questionnaires, identified the conceptual framework, and engaged a sample of the target population.

## Limitations

The CVR rating was not easy to understand by most children, especially the younger ones. Therefore, there were discrepancies between CVR ratings and what children mentioned during the interviews about how important each item was, which led to discrepancies with HCPs' CVR ratings for a few items. We analyzed the findings using multiple data sources to develop a comprehensive understanding about the face and content validity concerns and decide upon the inclusion of items in the modified questionnaires. Also, we used the percentage of agreement between groups of participants, which has helped to quantify the agreement between participating groups to ensure a rigorous assessment of content validity.

In Phase 2, children, parents, and oncology HCPs were recruited to determine the face and content validity of the WHAT. This was because we aimed to test the face and content validity of the WHAT by a sample of the populations who will use the WHAT in clinical settings: the respondents (children and parents), and data users (oncology HCPs who will use the data to

initiate clinical discussion). However, including complementary HCPs in future research would enrich our understanding about CHA use and enhance the content of the modified WHAT questionnaires. We were also not able to recruit participants from pediatric oncology newly arrived immigrant populations in Canada; thus, future research should be conducted to ask for their perspectives, and tailor the WHAT content to their CHA use experience. This is crucial as the WHAT could be used to enrich HCPs knowledge about the use of CHA by these importation populations. Finally, further research needs to be conducted in the context of pediatric oncology in low-middle-income countries before the WHAT can be utilized in these countries as clinical CHA questionnaires.

## Conclusions

We present the first steps of adapting and testing the face and content validity of two versions of clinical CHA questionnaires to be used in pediatric oncology. Children, parents, and HCPs agreed on the content of both versions of the WHAT questionnaires after one cycle of Delphi survey. Two iterative cycles of individual interviews and modifications made to the WHAT questionnaires according to suggestions from children, parents and HCPs have helped ensure adequate face and content validity for use in pediatric oncology. The next steps of the validation process will include determining the interrater reliability, construct validity, and the feasibility of administration of the modified versions of the WHAT questionnaires among children with cancer and their parents. Clinical utility testing of the modified WHAT is another planned step prior will be able to use it in clinical settings. The modified WHAT questionnaires will then help HCPs assess and document CHA use steadily and initiate communication and knowledge exchange about the benefits and risks of CHA between families and HCPs, which ultimately may enhance the disclosure rate, minimize the risk of potential CHA-conventional treatment interaction, and ensure informed decision making about using CHA. The routine use of the modified WHAT questionnaires may help HCPs and researchers to determine what approaches are being used and which may be helpful, thus leading to the development of a new CHA research agenda for pediatric oncology.

## Supporting information

**S1 Appendix. Conceptual model.**
(PDF)

**S2 Appendix. The child version of the modified WHAT questionnaire.**
(PDF)

**S3 Appendix. The parent version of the modified WHAT questionnaire.**
(PDF)

## Acknowledgments

The authors would like to thank the administrators and leaders of OPACC and SIOP Global Health Network for facilitating the recruitment of participants. We also thank Cynthia Nguyen for providing administrative support; and Rachel Hamilton, Grace Richandi, and Brittany Wiles for pilot testing the electronic surveys. We would also like to acknowledge Brittany Wiles for assistance in Phase 2 content analysis.

## Author Contributions

**Conceptualization:** Mohammad R. Alqudimat, Karine Toupin April, Lindsay Jibb, Charles Victor, Paul C. Nathan, Jennifer Stinson.

**Data curation:** Mohammad R. Alqudimat, Paul C. Nathan.

**Formal analysis:** Mohammad R. Alqudimat, Karine Toupin April, Lindsay Jibb, Jennifer Stinson.

**Funding acquisition:** Mohammad R. Alqudimat, Karine Toupin April, Lindsay Jibb, Charles Victor, Paul C. Nathan, Jennifer Stinson.

**Investigation:** Mohammad R. Alqudimat, Karine Toupin April, Lindsay Jibb, Jennifer Stinson.

**Methodology:** Mohammad R. Alqudimat, Karine Toupin April, Lindsay Jibb, Charles Victor, Paul C. Nathan, Jennifer Stinson.

**Project administration:** Mohammad R. Alqudimat, Jennifer Stinson.

**Resources:** Mohammad R. Alqudimat, Karine Toupin April, Lindsay Jibb, Charles Victor, Paul C. Nathan, Jennifer Stinson.

**Software:** Mohammad R. Alqudimat, Jennifer Stinson.

**Supervision:** Karine Toupin April, Lindsay Jibb, Charles Victor, Paul C. Nathan, Jennifer Stinson.

**Validation:** Mohammad R. Alqudimat, Karine Toupin April, Lindsay Jibb, Charles Victor, Paul C. Nathan, Jennifer Stinson.

**Visualization:** Mohammad R. Alqudimat.

**Writing – original draft:** Mohammad R. Alqudimat.

**Writing – review & editing:** Mohammad R. Alqudimat, Karine Toupin April, Lindsay Jibb, Charles Victor, Paul C. Nathan, Jennifer Stinson.

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
