## [Decision Letter · Decision Letter 0]

3 Jan 2023

PONE-D-22-31203How to assess and discuss complementary health approaches use in pediatric oncology: Adaptation and preliminary validation of the "Which Health Approaches and Treatments Are you Using?" (WHAT) questionnairesPLOS ONE

Dear Dr. Alqudimat,

Thank you for submitting your manuscript to PLOS ONE. After careful consideration, we feel that it has merit but does not fully meet PLOS ONE’s publication criteria as it currently stands. Therefore, we invite you to submit a revised version of the manuscript that addresses the points raised during the review process.

We look forward to receiving your revised manuscript.

Kind regards,

Erik Loeffen, M.D., Ph.D.

Academic Editor

PLOS ONE

Journal Requirements:

Reviewers' comments:

Reviewer's Responses to Questions

**Comments to the Author**

1. Is the manuscript technically sound, and do the data support the conclusions?

Reviewer #1: Yes

Reviewer #2: Yes

2. Has the statistical analysis been performed appropriately and rigorously? 

Reviewer #1: Yes

Reviewer #2: Yes

3. Have the authors made all data underlying the findings in their manuscript fully available?

Reviewer #1: Yes

Reviewer #2: Yes

4. Is the manuscript presented in an intelligible fashion and written in standard English?

Reviewer #1: No

Reviewer #2: Yes

5. Review Comments to the Author

Reviewer #1: A validated tool for the assessment of complementary health approaches by patients and their family is much needed. The project involved included international providers through the SIOP working group which is commended.

Title: The title is too long, suggestion below

Assessment of complementary health approaches in pediatric oncology: Modification of the Which Health Approaches and Treatment Are you Using (WHAT) questionnaire

Keywords: Too many

Abstract: The abstract should draw in the interest of the reader; this abstract is too wordy.

First sentence: Complementary health approaches (CHA) are commonly used by pediatric patients with cancer; however, few health care providers (HCP) inquire regarding the use of CHA for symptom alleviation.

Reframe from using we in the method and results of the abstract and in these sections of the manuscript

The manuscript is too long, needs editing and sentences are run on

Design

Suggest you delete: This article describes

In adapting and validating the WHAT questionnaire, a Delphi survey was conducted. This survey was distributed electronically to pediatric patients with cancer, parents, pediatric oncology HCPs, pediatric complementary care HCPs, and researchers. Phase 1 allowed consensus and adaption of the questionnaire domains and items for use in pediatric oncology. Phase 2 explored the face and content validity of the questionnaire, assessing relevance, comprehensiveness, and comprehension in the context of pediatric oncology.

Line 120: states semi-structured interview… using cognitive interviews. This sentence is not clear

Line 122: Interviews focused instead of were focused

Line 132 Participants Section

A convenience sample

Line 134-5 Patient eligibility included children diagnosed with cancer ≥ 3 months, English-speaking, 8-18 years of age, with a history of CHA during cancer therapy.

Line 139-143 describing HCP recruitment is not clear and too wordy. It is unclear who comprises “researchers”

Paragraph Procedure Line 161. “We” used 5 times in one paragraph and starts 3 of the sentences

On the following page, “we”

overused and distracting.

Line 173: Participants completed a demographic survey and were then asked to suggest modifications in the questionnaire domains and items, as well as recommendations for domains and items that might be added.

The data analysis should not include pronouns, just describe how the analysis was conducted

The result section is too long, and the authors have inferred opinions and comments that should not be included in a result section

For example, Line 277; These questions would be best for research purposes

Page 13 the first paragraph is too wordy and hard to follow

Line 256-258 needs editing for clarification

Line 261: The median rating by HCPs was 9 for all domains and approximately (the end of the sentence is not clear) they rated half the items at a 9?

Line 270: Only three participants suggested new domains… These suggestions included….

Line 295: All participants endorsed understanding the content of the question, expect two 10 year old children.

Line 335: results should not describe that redcap was used for the project

Line 435 speaks of two children 8 and 9, but previously they were 10. Are these different patients?

The results in many sections appear repetitive. Last paragraph page 25

Reviewer #2: Thank you for the opportunity to review this paper. In general, the manuscript is well-written and organized. The topic is important, and the questionnaire is a valuable tool to facilitate discussions among pediatric oncology patients, their parents, and healthcare providers.

I have some minor comments:

Pg 8. Line 146. The authors state they are targeting a response rate of 17%; however, they don’t provide the population size (e.g., the number of members of the organization or children at the Hospital from sick children). What was the denominator? Did they reach their 17% response rate target?

Pg11, line 219. The author states that MA and BW did the content analysis, but BW is not on the list of authors.

Pg 13. Lines 247-252 These percentages add to 99%; who was the other 1%? (The numbers add up, but the percentages do not)

Pg6, line 106 the sentence states: “Both the child- and parent proxy-report versions of the original WHAT questionnaires include four domains: child CAH use.” I believe the authors meant to write CHA; please double-check.

In table 2, under the communication domain, check spelling; now it reads; “communication about CAH” I think it is CHA.

Discussion:

The authors interviewed HCP from different regions of the world, but they don’t discuss how this tool can be used or the implications it would have in high vs. low-income countries with a difference in survival rates.

Also, if HCP from different countries were interviewed, why were only Canadian patients included in the study? Wouldn’t cultural differences affect the understanding and interpretations of the questions? This might be something worth addressing in the discussion. What about immigrant populations in Canada? Perhaps this is a limitation of the development of the questionnaires. I think this is important to address because, in your study, you argue that it is important to adapt and tailor questionnaires to enrich HCP knowledge about CHA, and you might be missing the input of an often-underserved population that HCP has little knowledge about.

Why were no complementary HCPs included in phase 2: testing face and content validity of the WHAT questionnaires? Could this be another limitation of the study?

6. PLOS authors have the option to publish the peer review history of their article (what does this mean?). If published, this will include your full peer review and any attached files.

Reviewer #1: No

Reviewer #2: No

---

## [Author Response · Author response to Decision Letter 0]

8 Feb 2023

The revised version of our manuscript has been formatted using PLOS ONE's style requirements. We have also added complete REB information, including the full name of the REB, the REB protocol numbers, and more details describing the obtained informed written and implied consent. The details in the manuscript have been alighted to the ethical statement. 

We have carefully reviewed the feedback and revised the manuscript accordingly. Please see below for an itemized response to the reviewers’ comments. We hope the paper is now in an acceptable form for publication in PLOS ONE.

All page and line numbers included in our responses below refer to the unmarked version of our manuscript without tracked changes (i.e., labeled “manuscript”)

Reviewers' comments:

Reviewer #1: 

A validated tool for the assessment of complementary health approaches by patients and their family is much needed. The project involved included international providers through the SIOP working group which is commended.

Title: 

The title is too long, suggestion below

Assessment of complementary health approaches in pediatric oncology: Modification of the Which Health Approaches and Treatment Are you Using (WHAT) questionnaire

Response: We appreciate your suggestion, and we considered changing the title accordingly. We condensed the first part of the title using your suggestion. For the second part, we replaced the word "Adaptation" with "Modification", but we kept the phrase "preliminary validation" since this paper is about both modifying and testing the validity of the WHAT questionnaires. The new title is "Assessment of complementary health approaches use in pediatric oncology: Modification and preliminary validation of the "Which Health Approaches and Treatments Are you Using?" (WHAT) questionnaires".

Keywords: Too many

Response: We reduced the number of keywords and kept the essential ones (i.e., six keywords).

Abstract:

The abstract should draw in the interest of the reader; this abstract is too wordy.

First sentence: Complementary health approaches (CHA) are commonly used by pediatric patients with cancer; however, few health care providers (HCP) inquire regarding the use of CHA for symptom alleviation.

Response: We followed the PLOS ONE submission guidelines and included 300 words in the original abstract. Indeed, we considered your comment and suggestion. We have revised the first sentence according to your suggestion. Also, the other parts of the abstract have been revised and condensed. The revised abstract now includes 289 words (see pages 2 and 3). 

Reframe from using we in the method and results of the abstract and in these sections of the manuscript

Response: Thank you for the comment. We used the active voice "as much as possible to create direct, clear, and concise sentences" (Section 4.13 of the APA publication manual 7th edition). However, we agree that we overused the active voice in our first submission. Thus, the abstract and the manuscript have been revised in line with your comment. 

The manuscript is too long, needs editing and sentences are run on

Response: To address this comment, we have substantially shortened our manuscript by 51 lines. This was the net reduction after condensing the method, streamlining the results, and adding further details to the limitation sections (see below for more information). We also improved the presentation of Table 2 (i.e., results of the Delphi survey) by re-grouping the findings. So, the domain ratings have been moved to be presented first, followed by item ratings. 

Design:

Suggest you delete: "This article describes" 

In adapting and validating the WHAT questionnaire, a Delphi survey was conducted. This survey was distributed electronically to pediatric patients with cancer, parents, pediatric oncology HCPs, pediatric complementary care HCPs, and researchers. Phase 1 allowed consensus and adaption of the questionnaire domains and items for use in pediatric oncology. Phase 2 explored the face and content validity of the questionnaire, assessing relevance, comprehensiveness, and comprehension in the context of pediatric oncology.

Response: As per the reviewer's suggestions, we have revised this paragraph accordingly (see lines 109-114). 

Line 120: states semi-structured interview… using cognitive interviews. This sentence is not clear

Response: To improve clarity, this sentence has been revised, and the phrase "using cognitive interviews" has been replaced with the phrase "using think-aloud and probing techniques" (lines 117-118) to clarify the two sub-types of cognitive interviewing methods that have been used in Phase 2. 

Line 122: Interviews focused instead of were focused

Response: As suggested, this sentence has been modified accordingly (line 118). 

Line 132 Participants Section

A convenience sample

Response: We have revised this statement (line 131). 

Line 134-5: Patient eligibility included children diagnosed with cancer ≥ 3 months, English-speaking, 8-18 years of age, with a history of CHA during cancer therapy.

Response: These lines have been revised following your suggestion (lines 132-135). We kept one part of the original sentence: "[…] having used at least one type of CHA since cancer diagnosis according to self- or proxy-report." to clarify how we identified the CHA users since this would not be a documented piece of information in the medical record. 

Line 139-143 describing HCP recruitment is not clear and too wordy. It is unclear who comprises "researchers"

Response: This part includes long names of two groups (i.e., SIOP and PedCAM) that were mentioned for the first time, making it wordy. However, those lines have been revised to enhance clarity (lines 137 to 142). The word "researchers" has been clarified by adding "CHA" to be "CHA researchers" (line 139). 

Paragraph Procedure Line 161. "We" used 5 times in one paragraph and starts 3 of the sentences

On the following page, "we" overused and distracting.

Response: We have modified the writing style for this procedure paragraph and for the following page. The text has been edited according to your comment (see pages 8 and 9), and sentences started with "We" have been minimized significantly throughout the manuscript. 

Line 173: Participants completed a demographic survey and were then asked to suggest modifications in the questionnaire domains and items, as well as recommendations for domains and items that might be added.

Response: This sentence has been modified according to your suggestion with a slight change to show the correct order of our procedure steps. Here is the edited sentence (lines 167 – 169): "participants were asked to suggest modifications in the domains and items, as well as recommendations for domains and items that might be added and were then asked to complete a demographic survey." 

The data analysis should not include pronouns, just describe how the analysis was conducted

Response: We revised the data analysis section accordingly. Pronouns have been removed and we shortened this section (see lines 170-172 and 201-219). 

The result section is too long, and the authors have inferred opinions and comments that should not be included in a result section

For example, Line 277; These questions would be best for research purposes

Response: We share your concern that the results section in the first submission was too long. The results section has been shortened significantly. 

Like previous studies on instrument validation and following COSMIN guidelines, we used qualitative data as evidence to guide our decisions about keeping, deleting, or modifying the WHAT items. For the Delphi study, quantitative data was described in the results section without including our opinion. For the short answers by participants that included suggestions for adding some domains, we analyzed these short responses and used them as evidence to decide whether to adjust the WHAT content accordingly. 

The analysis process for cognitive interviews included transcribing interviews, documenting field notes, summarizing participants' interpretations/understanding of items, identifying face and content validity concerns, and then deciding about each item (see data analysis paragraph on page 10). Also, we aimed to weigh the input from different data sources (i.e., triangulation) and make the decision for each item – whether to retain, omit, or revise. Thus, decision-making about items is considered the final step of our analysis, leading to the final results (i.e., the modified WHAT versions). Therefore, we included how we made the decision to modify the original WHAT in the results section. 

Page 13 the first paragraph is too wordy and hard to follow

Response: This paragraph has been edited to clarify the numbers and the proportions (see lines 234 – 238). Also, the CHA definition has been revised to improve the reading flow. 

Line 256-258 needs editing for clarification

Line 261: The median rating by HCPs was 9 for all domains and approximately (the end of the sentence is not clear) they rated half the items at a 9?

Response: We appreciate your suggestion. This section has been revised. All findings in the text that overlapped with information found in Table 2 have been removed (see lines 239 to 247). We also reorganized Table 2 to make it easier to follow. First, the ratings of the domains were presented, followed by the ratings of each item. 

Line 270: Only three participants suggested new domains… These suggestions included….

Response: We appreciate your suggestion. This line has been edited accordingly (lines 239 and 242). 

Line 295: All participants endorsed understanding the content of the question, expect two 10 year old children.

Response: Thank you for the suggestion. This line has been edited, and extra words have been removed (lines 262-263). 

Line 335: results should not describe that redcap was used for the project

Response: Agreed, we revised and moved this part about REDCap from the section about the results of the first cycle of Phase 2 to the "Resulting modified WHAT questionnaires" section (page 24). We wanted to highlight that switching from paper-based to electronic versions was done using a research tool (i.e. REDCap). Once the modified WHAT is ready to be used in clinical practice, it will ideally be integrated into an electronic medical record platform to be part of the clinical assessment. 

Line 435 speaks of two children 8 and 9, but previously they were 10. Are these different patients?

Response: Our response to this question is yes; these are different participants. The two 10 years old children participated in cycle 1 of Phase 2 (see line 262 under cycle 1 findings). In cycle 2, we had a new sample of participants (lines 363-364), of which two children 8 and 9 years old participated (line 377). 

The results in many sections appear repetitive. Last paragraph page 25

Response: The results sections have been revised and shortened substantially. 

Reviewer #2: 

Thank you for the opportunity to review this paper. In general, the manuscript is well-written and organized. The topic is important, and the questionnaire is a valuable tool to facilitate discussions among pediatric oncology patients, their parents, and healthcare providers. I have some minor comments.

Pg 8. Line 146. The authors state they are targeting a response rate of 17%; however, they don't provide the population size (e.g., the number of members of the organization or children at the Hospital from sick children). What was the denominator? Did they reach their 17% response rate target?

Response: We appreciate your suggestion. The response rate for each invited group has been added (lines 224-226): "The Delphi survey was completed by 61 participants with response rates per group as 68% (15/22) of children, 59% (13/22) of parents, 24% (10/41) of PedCAM pediatric HCPs and complementary HCPs and researchers, and 30% (23/77) of SIOP oncology HCPs." 

Pg11, line 219. The author states that MA and BW did the content analysis, but BW is not on the list of authors.

Response: Thank you for the comment. Yes, Brittany Wiles (BW) is not one of the authors. We meant to add the acknowledgement section in the first submission. Thus, the acknowledgment section has been added to the revised version (page 31) and we acknowledged BW efforts in the content analysis. 

Pg 13. Lines 247-252 These percentages add to 99%; who was the other 1%? (The numbers add up, but the percentages do not)

Response: We agree that this 1% has been missed by rounding up the percentages to zero decimal places. Thus, we re-rounded the numbers to one decimal place. Now, these percentages add to 100% (i.e., 80.3%, 16.4%, 3.3%); see lines 234-238. 

Pg6, line 106 the sentence states: "Both the child- and parent proxy-report versions of the original WHAT questionnaires include four domains: child CAH use." I believe the authors meant to write CHA; please double-check.

In table 2, under the communication domain, check spelling; now it reads; "communication about CAH" I think it is CHA.

Response: Yes, we meant to say "CHA". These typos have been corrected. 

Discussion:

The authors interviewed HCP from different regions of the world, but they don't discuss how this tool can be used or the implications it would have in high vs. low-income countries with a difference in survival rates.

What about immigrant populations in Canada? Perhaps this is a limitation of the development of the questionnaires. I think this is important to address because, in your study, you argue that it is important to adapt and tailor questionnaires to enrich HCP knowledge about CHA, and you might be missing the input of an often-underserved population that HCP has little knowledge about.

Response: We agree that testing the WHAT in low-middle-income countries and by a sample of pediatric oncology immigrants in Canada is a limitation of our study and requires future research. This limitation has been added to the limitation section (see lines 488 - 494).

Also, if HCP from different countries were interviewed, why were only Canadian patients included in the study? Wouldn't cultural differences affect the understanding and interpretations of the questions? This might be something worth addressing in the discussion. 

Response: We only invited international HCPs from the International Society of Pediatric Oncology to participate in a Delphi survey (no interviews) to get diverse perspectives and reach consensus on the domains and items of the WHAT questionnaires that meant to be used in the multi-cultural Canadian pediatric oncology populations. The Delphi survey is not a validity testing study; it is an important step in instrument development/adaptation (i.e., to identify the pool of domains and items that would be tested for validity and reliability). So, the first validity test for the WHAT was done in Phase 2, as we tested the content of the questionnaires in the Canadian context (i.e., no international participants).

The results of the Delphi survey of Canadian children/parents, and Canadian/International HCPs showed that all domains and items of the WHAT reached consensus after one cycle – this might highlight that the original WHAT included the key core set of domains and items that a sample of participants from different cultural backgrounds agreed to include after only one cycle of Delphi survey. Also, this might reflect that we included culturally diverse sample of Canadian children and parents in our study. We added some background details to Tables 1 and 2 to show the ethnic diversity of the children and parents who participated in Phases 1 and 2 (see pages 11-12 and 16-18).

Why were no complementary HCPs included in phase 2: testing face and content validity of the WHAT questionnaires? Could this be another limitation of the study?

Response: We agree that including complementary HCPs would enrich our understanding of the important items to include in the WHAT questionnaires. This point has been added to the limitations section (see lines 483-488).

---

## [Decision Letter · Decision Letter 1]

2 May 2023

PONE-D-22-31203R1Assessment of complementary health approaches use in pediatric oncology: Modification and preliminary validation of the "Which Health Approaches and Treatments Are You Using?" (WHAT) questionnairesPLOS ONE

Dear Dr. Alqudimat,

Thank you for submitting your manuscript to PLOS ONE. After careful consideration, we feel that it has merit but does not fully meet PLOS ONE’s publication criteria as it currently stands. Therefore, we invite you to submit a revised version of the manuscript that addresses the points raised during the review process.

We look forward to receiving your revised manuscript.

Kind regards,

Erik Loeffen, M.D., Ph.D.

Academic Editor

PLOS ONE

Journal Requirements:

Additional Editor Comments:

First of all I want to congratulate the authors with an interesting an well-conducted study. The authors have incorporated the good suggestions from the reviewers, and as a result of this the paper has improved substantially. Content-wise this paper is ready for acceptance and publication in Plos ONE. Nevertheless before I make a decision to accept I would urge the authors to shorten the manuscript substantially. From what I can see the text body is between 5000 and 6000 words, and although Plos ONE has no strict word limit this feels much too long for a Delphi&interview/validation study. For instance, line 279-300 is essentially "multiple participants did not find the paper format and the use of tables user-friendly, therefore we chose to adjust...". All the results and numbers can either be removed, put into a table or (my preference) put in detail in the supplementary data. I think shortening the manuscript would increase readability in a great manner. (Especially given the nice, large tables there is no need for an abundance of results/numbers in the text.)

For a swift proces, at resubmission I will handle the manuscript personally, sending for peer review would then not be necessary (as we have completed the step already).

Reviewers' comments:

Reviewer's Responses to Questions

**Comments to the Author**

1. If the authors have adequately addressed your comments raised in a previous round of review and you feel that this manuscript is now acceptable for publication, you may indicate that here to bypass the “Comments to the Author” section, enter your conflict of interest statement in the “Confidential to Editor” section, and submit your "Accept" recommendation.

Reviewer #1: All comments have been addressed

Reviewer #2: All comments have been addressed

2. Is the manuscript technically sound, and do the data support the conclusions?

Reviewer #1: Yes

Reviewer #2: Yes

3. Has the statistical analysis been performed appropriately and rigorously? 

Reviewer #1: Yes

Reviewer #2: Yes

4. Have the authors made all data underlying the findings in their manuscript fully available?

Reviewer #1: Yes

Reviewer #2: Yes

5. Is the manuscript presented in an intelligible fashion and written in standard English?

Reviewer #1: (No Response)

Reviewer #2: Yes

6. Review Comments to the Author

Reviewer #1: (No Response)

Reviewer #2: Thank you for addressing all my comments. All the comments were appropriately answered and the included in the text.

7. PLOS authors have the option to publish the peer review history of their article (what does this mean?). If published, this will include your full peer review and any attached files.

Reviewer #1: No

Reviewer #2: No

---

## [Author Response · Author response to Decision Letter 1]

12 Jun 2023

Dr. Erik Loeffen

Academic Editor

PLOS ONE

June 12, 2023

Dear Dr. Loeffen, 

Thank you for the constructive feedback on our manuscript “Assessment of complementary health approaches use in pediatric oncology: Modification and preliminary validation of the “Which Health Approaches and Treatments Are You Using?” (WHAT) questionnaires. 

We have carefully reviewed your comments and revised the manuscript accordingly. Please see below for an itemized response to your comments. We hope the paper is now in an acceptable form for publication in PLOS ONE. Thank you for the opportunity to revise and resubmit our manuscript.

Sincerely, 

Mohammad R. Alqudimat (Corresponding Author)

Ph.D. Candidate - University of Toronto

m.alqudimat@mail.utoronto.ca

Thank you for the comments and suggestions. We have substantially shortened our manuscript by 89 lines (1,339 words)—the revised text body is now 4,491 words. The following is a summary of this revision. All page and line numbers included in our response below refer to the unmarked version of our manuscript without tracked changes (labelled “manuscript”): 

1) We have condensed the results section, as suggested, and moved the details to the existing Table 1 and two new Tables 4 and 6. 

a. The details about participants’ geographical locations have been removed from the first paragraph of the results (page 11) and moved to Table 1 “Characteristics of participants who participated in the Delphi survey” (page 12). 

b. The findings of the first cycle of the face and content validity study have been condensed, and specific details about comprehensiveness, comprehensibility, and relevance have been moved to a new Table 4 “Findings of the first cycle of the individual interviews regarding the comprehensiveness, comprehensibility, and relevance” (page 18). Only high-level findings have been kept in the text body (page 17). 

c. The section titled “Resulting modified WHAT questionnaires” has been condensed to describe the modified WHAT. 

d. The findings of the second cycle of the face and content validity study have been condensed, and details about comprehensiveness, comprehensibility, and relevance were moved to a new Table 6 “Findings of the second cycle of the individual interviews regarding the comprehensiveness, comprehensibility, and relevance” (page 22). Only high-level findings were kept in the text body (page 21).

2) Minor changes to parts of the design, discussion, limitation, and conclusion have been made. We have condensed or clarified these parts, and no new ideas have been added. 

a. The procedure of Phase 1: 

i. Lines 160-161: The sentence “Children and parents were asked to rate the importance of each domain and item of the child self-report and parent proxy-report versions, respectively” has been changed to “Children were asked to rate the importance of each domain and item of the self-report version, and parents were asked to rate the proxy-report.”

b. The procedure and data analysis of Phase 2: 

i. Lines 191-197: Three sentences describing the interview procedure have been reorganized to ease the readability and to reduce the word count. Here is the revised part: “During the interviews, children were asked to read over the child self-report, parents read the parent proxy-report. Children and parents were then asked to describe what they understood about the meaning of the item and whether it was easy to understand (comprehensibility), any essential items were missing (comprehensiveness), and all items and response options were relevant to the questionnaires’ purpose. HCPs were asked to review both versions of the WHAT questionnaires and to answer open-ended questions about item relevance and comprehensiveness of the questionnaires.”

ii. Lines 217-218: The word “retaining” has been clarified by adding two words, “modifying, removing, or retaining”. 

c. Discussion: 

i. Lines 309-311: The sentence “We introduce the first preliminary face and content validity tested electronic disease-specific CHA questionnaires (i.e., the modified WHAT) that are designed to assess CHA use and initiate clinical dialogue about CHA use in pediatric oncology” has been condensed and simplified to be “We introduce the first electronic disease-specific CHA questionnaires (i.e., the modified WHAT) that have evidence of face and content validity and are designed to assess CHA use and initiate clinical discussions about CHA in pediatric oncology.”

ii. Lines 370-372: The sentence has been clarified by adding “Agreement between the self- and proxy-report is not granted unless confirmed by inter-rater reliability testing; for example,” to the original sentence “previous research has shown that proxy-report results are not equivalent to the self-report in a study on perceptions of quality of life in children under chiropractic care.”

d. Limitations: 

i. Lines 392-393: Minor changes to clarify the original two sentences have been made. Here are the edited sentences “In Phase 2, children, parents, and oncology HCPs were recruited to determine the face and content validity of the WHAT. This was because we aimed to test the face and content validity of the WHAT by a sample of the populations who will use the WHAT in clinical settings: the respondents (children and parents), and data users (oncology HCPs who will use the data to initiate clinical discussion).”

ii. Lines 398-401: A few words were replaced to clarify one study limitation. Here is the edited sentence “We were also not able to recruit participants from pediatric oncology newly arrived immigrant populations in Canada; thus, future research should be conducted to ask for their perspectives, and tailor the WHAT content to their CHA use experience.”

e. Conclusions:

i. Lines 407-410: Two sentences have been edited to clarify and simplify the points. The edited sentences are shown as “Children, parents, and HCPs agreed on the content of both versions of the WHAT questionnaires after one cycle of Delphi survey. Two iterative cycles of individual interviews and modifications made to the WHAT questionnaires according to suggestions from children, parents and HCPs have helped ensure adequate face and content validity for use in pediatric oncology.”

3) No changes to the reference list have been made. 

4) The revised version of our manuscript has been formatted using PLOS ONE’s style requirements.

---

## [Decision Letter · Decision Letter 2]

2 Nov 2023

Assessment of complementary health approaches use in pediatric oncology: Modification and preliminary validation of the "Which Health Approaches and Treatments Are You Using?" (WHAT) questionnaires

PONE-D-22-31203R2

Dear Dr. Alqudimat,

We’re pleased to inform you that your manuscript has been judged scientifically suitable for publication and will be formally accepted for publication once it meets all outstanding technical requirements.

Kind regards,

Marianne Clemence

Staff Editor

PLOS ONE

Additional Editor Comments (optional):

Reviewers' comments:

Reviewer's Responses to Questions

**Comments to the Author**

1. If the authors have adequately addressed your comments raised in a previous round of review and you feel that this manuscript is now acceptable for publication, you may indicate that here to bypass the “Comments to the Author” section, enter your conflict of interest statement in the “Confidential to Editor” section, and submit your "Accept" recommendation.

Reviewer #1: All comments have been addressed

Reviewer #2: All comments have been addressed

2. Is the manuscript technically sound, and do the data support the conclusions?

Reviewer #1: Yes

Reviewer #2: Yes

3. Has the statistical analysis been performed appropriately and rigorously? 

Reviewer #1: Yes

Reviewer #2: Yes

4. Have the authors made all data underlying the findings in their manuscript fully available?

Reviewer #1: Yes

Reviewer #2: Yes

5. Is the manuscript presented in an intelligible fashion and written in standard English?

Reviewer #1: Yes

Reviewer #2: Yes

6. Review Comments to the Author

Reviewer #1: (No Response)

Reviewer #2: (No Response)

7. PLOS authors have the option to publish the peer review history of their article (what does this mean?). If published, this will include your full peer review and any attached files.

Reviewer #1: No

Reviewer #2: No

---

## [Editor Report · Acceptance letter]

24 Feb 2024

PONE-D-22-31203R2 

PLOS ONE

Dear Dr. Alqudimat, 

I'm pleased to inform you that your manuscript has been deemed suitable for publication in PLOS ONE. Congratulations! Your manuscript is now being handed over to our production team.

Kind regards, 

on behalf of

Dr Marianne Clemence 

Staff Editor

PLOS ONE